# Targeted Endoradiotherapy with Lu_2_O_3_-iPSMA/-iFAP Nanoparticles Activated by Neutron Irradiation: Preclinical Evaluation and First Patient Image

**DOI:** 10.3390/pharmaceutics14040720

**Published:** 2022-03-27

**Authors:** Myrna Luna-Gutiérrez, Blanca Ocampo-García, Nallely Jiménez-Mancilla, Alejandra Ancira-Cortez, Diana Trujillo-Benítez, Tania Hernández-Jiménez, Gerardo Ramírez-Nava, Rodrigo Hernández-Ramírez, Clara Santos-Cuevas, Guillermina Ferro-Flores

**Affiliations:** 1Department of Radioactive Materials, Instituto Nacional de Investigaciones Nucleares, Ocoyoacac 52750, Mexico; myrna.luna@inin.gob.mx (M.L.-G.); blanca.ocampo@inin.gob.mx (B.O.-G.); alejandra.a.servicios@inin.gob.mx (A.A.-C.); sarahitrujillo097@hotmail.com (D.T.-B.); tania.hernandez@inin.gob.mx (T.H.-J.); gerardo.r.servicios@inin.gob.mx (G.R.-N.); 2Cátedras CONACyT, Instituto Nacional de Investigaciones Nucleares, Ocoyoacac 52750, Mexico; nallely.jimenez@inin.gob.mx; 3Nuclear Medicine Department, Hospital Médica Sur, Mexico City 14080, Mexico; rhernandezr@medicasur.org.mx

**Keywords:** lutetium oxide nanoparticles, prostate-specific membrane antigen inhibitor, fibroblast activation protein inhibitor, lutetium-177

## Abstract

Prostate-specific membrane antigen (PSMA) is expressed in a variety of cancer cells, while the fibroblast activation protein (FAP) is expressed in the microenvironment of tumors. Previously, we reported the ability of iPSMA and iFAP ligands to specifically target PSMA and FAP proteins, as well as the preparation of stable ^177^Lu_2_O_3_ nanoparticles (<100 nm) functionalized with target-specific peptides. This research aimed to evaluate the dosimetry and therapeutic response of Lu_2_O_3_-iPSMA and Lu_2_O_3_-iFAP nanoparticles activated by neutron irradiation to demonstrate their potential for theranostic applications in nuclear medicine. The biokinetic behavior, radiation absorbed dose, and metabolic activity ([^18^F]FDG/micro-PET, SUV) in preclinical tumor tissues (athymic mice), following treatment with ^177^Lu_2_O_3_-iPSMA, ^177^Lu_2_O_3_-iFAP or ^177^Lu_2_O_3_ nanoparticles, were assessed. One patient with multiple colorectal liver metastases (PSMA-positive) received ^177^Lu_2_O_3_-iPSMA under a “compassionate use” protocol. Results indicated no significant difference (*p* < 0.05) between ^177^Lu_2_O_3_-iPSMA and ^177^Lu_2_O_3_-iFAP, regarding tumor radiation absorbed doses (105 ± 14 Gy, 99 ± 12 Gy and 58 ± 7 Gy for ^177^Lu_2_O_3_-iPSMA, ^177^Lu_2_O_3_-iFAP, and ^177^Lu_2_O_3_, respectively) and tumor metabolic activity (SUV of 0.421 ± 0.092, 0.375 ± 0.104 and 1.821 ± 0.891 for ^177^Lu_2_O_3_-iPSMA, ^177^Lu_2_O_3_-iFAP, and ^177^Lu_2_O_3_, respectively) in mice after treatment, which correlated with the observed therapeutic response. ^177^Lu_2_O_3_-iPSMA and ^177^Lu_2_O_3_-iFAP significantly inhibited tumor progression, due to the prolonged tumor retention and a combination of ^177^Lu radiotherapy and iPSMA or iFAP molecular recognition. There were negligible uptake values in non-target tissues and no evidence of liver and renal toxicity. The doses received by the patient’s liver metastases (42–210 Gy) demonstrated the potential of ^177^Lu_2_O_3_-iPSMA for treating colorectal liver metastases.

## 1. Introduction

Research on nanoparticle-based radiopharmaceuticals for the treatment of various types of cancer is of interest in the field of human health, due to the high tumor retention and high internalization in cancer cells of previously-reported targeted radionanosystems [1,2,3,4].

Lutetium-177 has been widely used as the radionuclide of choice for peptide-targeted endoradiotherapy [5,6]. During the last decade many studies have reported the preclinical evaluation of polymeric and metallic nanoparticles labeled with ^177^Lu. However, the literature related to the preparation of lutetium nanoparticles is scarce and focused on preparing luminescent doped systems (Lu_2_O_3_: Ln^3+^, Ln = Eu, Er, Yb) or contrast agents [7,8,9,10].

Fibroblast activation protein (FAP) is expressed mainly in activated stromal fibroblasts, present in the tumor microenvironment of most human epithelial tumors, but not on the cell membrane of normal fibroblasts [11].

Prostate-specific membrane antigen (PSMA) is a protein that is overexpressed in 90% of advanced prostate tumors and the neovasculature of the tumor microenvironment of many types of advanced cancer, such as metastatic colon cancer, triple-negative breast cancer, and hepatocarcinoma, among others [12,13,14,15,16].

Previously, our research group reported the synthesis of lutetium sesquioxide nanoparticles functionalized with peptides and activated by neutron irradiation [17,18]. The chemical characterization of the nanosystems showed a well-defined quasi-spherical morphology with a monodisperse and monomodal distribution size (30–45 nm), and displayed characteristic radioluminescent properties. The radioluminescent property is due to the interaction between lutetium atoms and their emitted ionizing radiation. In addition, the Lu electronic configuration (4f^14^ 5d^1^ 6s^2^) allows the formation of sesquioxides with trivalent ions and the presence of f-f transitions between electrons of 4f orbitals shielded by 5s and 5p electrons, which results in photonic emissions (radioluminescence) with long half-lives and resistance to photobleaching.

The ^177^Lu_2_O_3_-iPSMA nanosystem demonstrated in vivo stability and affinity (K_d_ = 5.7 nM) for the PSMA protein, while ^177^Lu_2_O_3_-iFAP showed significant membrane binding in stromal cells expressing FAP receptors [17,19].

This research aimed to evaluate the biokinetics, dosimetry and preclinical therapeutic response of Lu_2_O_3_-iPSMA and Lu_2_O_3_-iFAP nanoparticles activated by neutron irradiation to demonstrate their potential for theranostic applications in nuclear medicine. As a proof of concept, the biokinetics and dosimetry of ^177^Lu_2_O_3_-iPSMA in liver metastases (PSMA-positive) of a patient with advanced colorectal cancer, was also assessed.

## 2. Materials and Methods

### 2.1. Synthesis of Lutetium Nanoparticles

Lutetium sesquioxide nanoparticles (Lu_2_O_3_ NPs) were synthesized by a precipitation-calcination method, as previously reported [17]. Briefly, a 10-mM solution of LuCl_3_ (anhydrous; powder; 99.99% trace metals basis; Sigma-Aldrich; Saint Louis, MO, USA) was prepared, onto which a 2-M (NH_4_)_2_CO_3_/NH_4_OH 1:1 (*v/v*) mixture was added dropwise until the solution reached pH 9, appearing as a fluffy white precipitate. Stirring was continued for an additional hour. The precipitate was washed with type-I water, employing several centrifugations at 2500× *g*/30 min, until the washing solution reached pH 7. Subsequently, the product was dried and calcined at 1000 °C for 24 h. The obtained powder was analyzed by transmission electron microscopy (TEM) (JEOL JEM 2010 HT, JEOL Inc, Peabody, MA, USA) and infrared spectroscopy (FT-IR, ATR platform) (PerkinElmer 2000, PerkinElmer, Waltham, MA, USA). Dynamic light scattering (DLS: Nanotrac wave, Model MN401) was also applied to verify the hydrodynamic nanoparticle size.

### 2.2. Preparation of ^177^Lu_2_O_3_-iPSMA and ^177^Lu_2_O_3_-iFAP Nanoparticles

The ^177^Lu_2_O_3_ nanoparticles were produced by neutron capture. For neutron activation, 12 mg of Lu_2_O_3_ nanoparticles were irradiated in the Triga MARK III nuclear reactor of the National Institute for Nuclear Research, Mexico (ININ, Ocoyoacac, Mexico), in two cycles of 20 h each, at a neutron flux of 3 × 10^13^ n/s^.^cm^2^, with an interval of 5 d between the two cycles.

Twenty-four hours after irradiation, the nanoparticles were dispersed in 10 mL of a 19-mM sodium citrate solution. Subsequently, 50 µL (0.1 mM) of the iPSMA peptide (ligand targeting PSMA) [20] or iFAP (ligand targeting FAP) [19] were added. The solution was divided into 1.5-mL volumes, which were added to sterile and apyrogenic 13-mL vials containing 3.5 mL of injectable-grade water. The vials were sealed with aluminum caps. The mixture was vigorously shaken, and the activity was verified in an activimeter (Capintec CRC-55tR). The radionanoparticles were sterilized in an autoclave at 121 °C and 21 psi for 20 min. The final colloidal solutions were subjected to bacterial endotoxin and sterility tests (pharmacopeial methods). Radiochemical purity was verified by ultracentrifugation (2500× *g* for 0.5 h; 30,000 MW cut-off; Amicon Ultra; Millipore: Burlington, MA, USA), in which functionalized [^177^Lu]Lu_2_O_3_ (^177^Lu_2_O_3_) nanoparticles remained in the filter, while [^177^Lu]LuCl_3_ and ^177^Lu_2_O_3_ (not nanoparticles) passed through the filter. The processes were carried out in areas with GMP certification. After completing radioactive decay, one sample was used to verify nanoparticle size distribution by TEM and DLS, as well as nanoparticle functionalization, via IR spectroscopy.

### 2.3. Cell Culture

The human colorectal cancer cell line HCT116 (ATCC^®^ CCL-247™) was cultured in RPMI-1640 (ATCC^®^ 30-2001™) medium (ATCC, Manassas, VA, USA). The medium was supplemented with 10% fetal bovine serum and 1% penicillin/streptomycin solution. Cells were grown under a 5% carbon dioxide atmosphere at 37 °C.

### 2.4. Bidistribution of ^177^Lu_2_O_3_-iPSMA and ^177^Lu_2_O_3_-iFAP Nanoparticles

In vivo studies in healthy mice (CD1 male mice; 8 weeks old; weight from 40 to 42 g) were performed in accordance with ethical regulations and standards for the handling of laboratory animals (Official Mexican Norm NOM-062-ZOO-1999). CD1 mice were injected, via the tail vein, with ^177^Lu_2_O_3_-iPSMA or ^177^Lu_2_O_3_-iFAP nanoparticles (50 μL; 13 MBq). Subsequently, whole-body radioluminescent images were obtained from three mice via a system equipped with a CCD camera (XTREME Imaging System; Bruker, Billerica, MA, USA), at the following post-injection acquisition times: 0.5, 1, 2, 3, 24, 48, 72 and 96 h. The other CD1 mice were sacrificed at the time intervals of 3, 48, 72 and 96 h (*n* = 3 for each time point) for each radionanosystem. The following organs were dissected: liver, spleen, lungs, pancreas, heart, and kidneys, whose activity was measured in a thallium-doped NaI detector, in order to determine the percentage of the dose injected per organ (%ID) with regard to the total activity initially injected. Two groups of healthy mice (*n* = 3) (CD1 mice; average weight of 40 g) were injected intravenously with a single dose (12 µg/40 g, 2 MBq) of ^177^Lu_2_O_3_-iPSMA or ^177^Lu_2_O_3_-iFAP nanoparticles and after 2 weeks blood samples were obtained for creatinine and liver enzyme quantitation as described below.

### 2.5. Bidistribution of Nanoparticles in Mice Bearing HCT116 Tumors

Athymic mice (nude male mice; 7 weeks old; weight from 20 to 22 g) were used for tumor induction by subcutaneous injection of HCT116 cells (1 million cells suspended in 0.2 mL PBS) into the leg muscle.

### 2.6. Therapeutic Protocol

Nude mice with HCT116-induced tumors (tumor size of 0.06 ± 0.01 g) were divided into 4 groups (*n* = 5), to receive one of the following radionanoparticle treatments (1–2 MBq/0.05 mL), via intratumoral injection, on days 1, 7 and 14: (a) ^177^Lu_2_O_3_-iPSMA, (b) ^177^Lu_2_O_3_-iFAP, and (c) ^177^Lu_2_O_3_. The fourth untreated group was used as a control. Each week, the width (*a*) and length (*L*) of tumors were measured with calipers, in order to determine the volume, using the Equation (1):(1)V=π6×(L)×(a2)

After 21 days, the tumor’s metabolic activity was assessed by [^18^F]FDG (2-deoxy-2-[^18^F]fluoro-D-glucose) PET/CT imaging. Finally, mice were sacrificed to draw blood samples for quantitation of creatinine and liver enzymes. The radiation absorbed dose assessment was performed by calculating the total number of nuclear transformations (*N*) that occurred in the tumors through the mathematical integration of the activity functions, *A_h_*(*t*), [$A_{h}\left(t\right)=q_{h}\left(t\right)e^{-\lambda_{Lu-177}t}$], considering the three injected activities (Equation (2)):(2)$$N_{TUMOR}=\int_{t=0}^{t=21\;d}A_{h}\left(t\right)dt+\int_{t=7\;d}^{t=21\;d}A_{h}\left(t\right)dt+\int_{t=14\;d}^{t=21\;d}A_{h}\left(t\right)dt$$

The *N* values were used with the sphere model included in the OLINDA/EXM code [21], to calculate the radiation doses to the tumors throughout the treatment.

### 2.7. Metabolic Activity Evaluation

[^18^F]FDG PET/CT images were obtained using a micro-SPECT/PET/CT equipment (Albira; ONCOVISION; Valencia, Spain). Mice included in the treatment protocol (at day 21) were anesthetized with 2% isoflurane and injected in the caudal vein with 5.55 MBq (0.05 mL) of [^18^F]FDG for whole-body imaging at 1 h post-injection. The PET field-of-view was 60 mm and the acquisition time was 7.5 min. The CT images were acquired with a current of 700 µA, a 35-kV sure voltage, and 600 projections. The mean standardized uptake value [mean SUV] was calculated using PMOD Data Analysis software.

### 2.8. Creatinine and Liver Enzyme Quantitation

The blood samples of mice, obtained at the end of each treatment, were used for creatinine, aspartate aminotransferase (AST), alanine aminotransferase (ALT), and lactate dehydrogenase (LDH) quantitation. Creatinine was evaluated titrimetrically using the picrate method. AST, ALT, and LDH were quantitated using a Roche Diagnostics kit for UV assays.

### 2.9. Clinical Study

A 67-year-old woman (weight of 57 kg; height of 162 cm), diagnosed with unresectable liver metastases from colorectal cancer, was enrolled in this study. Three months before enrollment, the patient underwent ultrasound and contrast-enhanced CT of the abdomen, which showed data of advanced metastatic colorectal cancer. [^18^F]FGD PET/CT, and colonoscopy with biopsy and molecular markers were employed, confirming moderately-differentiated intestinal-type invasive adenocarcinoma (stage IV), mutated KRAS (negative to BRAF, PIK3CA, and AKT1, NRAS mutation), negative for microsatellite instability, and with expression of the four proteins of the DNA repair genes. The primary tumor was located in the right iliac fossa, at the level of the ileocecal valve. The patient presented multiple liver metastases, with the presence of two large necrotic lesions; the largest was in segment IV, with a diameter of approximately 6 cm on the largest axis, which compressed the confluence of the right and left biliary branches and the common hepatic artery, causing severe dilatation of the intrahepatic bile duct. Through an endoscopic retrograde cholangiopancreatography procedure, it was possible to place a stent to drain the left side of the liver, but it was not possible to place a second stent for the right side, due to the obstruction caused by the large necrotic lesion. After resection of the primary colorectal tumor, a process of cholangitis was developed and treated with beta-lactam antibiotics. Subsequently, two external catheters were placed to drain the right side of the liver. The patient was intolerant to Folfox chemotherapy (oxyplatin 70 mg + folinic acid 400 mg + 5-FU 1800 mg), which generated severe hepatomegaly, generalized edema, and a state of prostration. A month and a half after the application of chemotherapy, a compassionate ^177^Lu_2_O_3_-iPSMA protocol was approved (Ethics Committee: Protocol No. 2021-MN01) as a possible alternative for the treatment of liver metastases. The patient signed a consent form after receiving detailed information regarding the aims of the study and agreed to the collection of data.

At the time of ^177^Lu_2_O_3_-iPSMA application, the patient had a leukocyte count of 8.1 × 10^9^/L, platelet count of 456 × 10^9^/L, hemoglobin level of 13.6 g/dL, serum creatinine level of 0.6 g/L, serum bilirubin level of 2.2 g/L (direct bilirubin of 1.7 g/L), serum albumin level of 30 g/L, AST level of 189.0 U/L, ALT level of 135.0 U/L, LDH level of 318.0 U/L, serum gamma glutamyltransferase level of 2666.0 U/L, and serum alkaline phosphatase level of 3413.0 U/L.

^99m^Tc-iPSMA (740 MBq) SPECT/CT imaging of the patient was obtained to evaluate PSMA expression. Afterwards, a dose of 185 MBq of ^177^Lu_2_O_3_-iPSMA nanoparticles was intravenously administered. A hybrid planar/SPECT (2D/3D) quantitation method was applied to obtain biokinetic data, as previously reported [3]. In brief, planar and SPECT/CT images were obtained at 1 h after ^177^Lu_2_O_3_-iPSMA administration, with a dual-head gamma camera (SPECT/CT; Siemens, Munich, Germany). Correction factors (*CF_Hybrid_*) between imaging modalities were calculated by dividing the activity in the organ of interest quantified by SPECT (reconstructions in kBq/mL) and the activity measured in the organ of interest on planar images. Subsequently, planar images were obtained at 24, 90 and 134 h. CF_Hybrid_ was applied in the quantitation of the planar method (*A*(*t*)*_P_*), to determine the volumetric activity (*A*(*t*)*_VOI_*), as follows: *A*(*t*)*_VOI_* = *CF_Hybrid_ × A*(*t*)*_P_*. The fractions of the injected activity in each source organ, including tumors (modeled as geometries of isolated spheres), were fitted to exponential models and radiation absorbed doses were calculated by using the OLINDA/EXM code [21].

### 2.10. Statistics

Differences among the preclinical treatment groups were analyzed by three-way ANOVA multiple comparisons (Tukey’s Multiple Comparisons Test; alpha of 0.05).

## 3. Results

TEM, DLS, and IR analyses of the nanoparticles corroborated the nanometric size (36 nm ± 7 nm by TEM and 105 nm ± 20 nm of hydrodynamic diameter determined by DLS), and the presence of a band at 575 cm^−1^, corresponding to the Lu–O stretching vibration, which is also proof of the correct crystallization and nanometric scale, as previously reported [17,18]. After neutron irradiation cycles and 24 h of decay, the specific activity of [^177^Lu]Lu_2_O_3_ nanoparticles was 843 MBq/mg. Sterile and apyrogenic colloidal solutions of [^177^Lu]Lu_2_O_3-_iPSMA (^177^Lu_2_O_3-_iPSMA) and [^177^Lu]Lu_2_O_3_-iFAP (^177^Lu_2_O_3-_iFAP) were obtained as pharmaceutical preparations, containing 1350 MBq in 5 mL of 6-mM sodium citrate solution (Figure 1). After complete decay, Lu_2_O_3-_iPSMA and Lu_2_O_3_-iFAP kept their monodispersed and monomodal distribution behavior and morphological characteristics, determined by DLS and TEM. IR spectra of the functionalized nanoparticles also showed the band at 575 cm^−1^ and a blue shift of most iFAP and iPSMA characteristic bands (20–100 cm^−1^), due to the formation of covalent bonds (Lu-DOTA) and changes in the way dipoles vibrate when interacting on the nanoparticle surface, regarding the free ligand.

As expected, the biodistribution profile of ^177^Lu_2_O_3_-iPSMA and ^177^Lu_2_O_3_-iFAP, after intravenous injection into healthy CD1 mice, indicated a considerable hepatic uptake for both nanosystems, without significant differences between them (*p* < 0.05) (Table 1 and Table 2). The second and third target organ was the spleen and lung, with approximately 15 and 70 times less uptake of nanoparticles than the liver at 3 h, respectively. Radioluminescent images of ^177^Lu_2_O_3_-iPSMA in healthy mice, at different times, also corroborated the liver as the primary target and source organ (Figure 2). Therefore, the intravenous administration of the radionanosystems for targeted treatment of liver metastases or hepatocarcinoma could be possible, since radiotherapy is usually indicated when a significant and inoperable tumor burden affects the hepatic tissue, as long as the nanoparticles are captured and retained in the tumor lesions.

The biokinetic evaluation of ^177^Lu_2_O_3_-iPSMA (Table 3) and ^177^Lu_2_O_3_-iFAP (Table 4) in nude mice bearing HCT116 tumors showed a statistically significant (*p* < 0.05) higher tumor retention of both nanosystems after intratumoral injection with regard to ^177^Lu_2_O_3_ nanoparticles (Table 5). The mathematical integration of the fit data as exponential models (biokinetic models) until ^177^Lu complete decay, showed total nuclear transformation values (N) of 597,600 ± 55,576 Bq^.^s for ^177^Lu_2_O_3_-iPSMA and 586,800 ± 49,878 Bq^.^s for ^177^Lu_2_O_3_-iFAP, which were significantly (*p* < 0.05) greater compared to those produced by ^177^Lu_2_O_3_ (339,840 ± 15,632 Bq^.^s). The differences in the percentage of injected dose retained in the tumors and *N* values are attributed to the target-specific recognition of iPSMA and iFAP ligands present in the functionalized nanosystems. Some uptake also occurred mainly in the liver and, to a lesser degree, in the spleen and kidney, with practically-negligible retention in other organs.

The tumor progression in all mice treated with ^177^Lu_2_O_3_ nanoparticles was significantly inhibited (*p* < 0.05) with regard to the control group (Figure 3). At 21 d, tumor size in the ^177^Lu_2_O_3_-iPSMA and ^177^Lu_2_O_3_-iFAP groups was 7.5 times smaller than that of the controls and three-fold smaller than in the ^177^Lu_2_O_3_ group (Figure 3). The tumor size progression correlated with the doses delivered to the HCT116 tumors in the order: ^177^Lu_2_O_3_-iFAP (105 ± 14 Gy), ^177^Lu_2_O_3_-iPSMA (99 ± 12 Gy), and ^177^Lu_2_O_3_ (58 ± 7 Gy) nanoparticles, although without significant (*p* < 0.05) difference between the functionalized nanoparticles (Figure 3).

Significant uptake of [^18^F]FDG in tumors produces high SUV values, indicating high metabolic activity in the viable cells of tumors. As shown in Table 6, the SUV values of the groups of mice treated with nanoparticles were significantly lower with regard to the control group (*p* < 0.05), while the SUV of the ^177^Lu_2_O_3_ group was significantly greater than that of the ^177^Lu_2_O_3_-iPSMA and ^177^Lu_2_O_3_-iFAP groups (*p* < 0.05).

As is known, AST and ALT significantly increase their enzymatic activity after a toxic exposure that affects the integrity of liver cells, while LDH is a stable cytosolic enzyme that is released after cytotoxic damage. Also, elevated creatinine levels indicate impaired kidney function. As shown in Table 6, no significant differences (*p* < 0.05) in the creatinine, AST, ALT, and LDH values between mice treated with radionanoparticles and those of the control group were found. The same behavior was observed in healthy mice (CD1 mice; average weight of 40 g) at 2 weeks after intravenous injection of the functionalized radionanoparticles, with a single dose of 12 µg/40 g (2 MBq), which indicated that the ^177^Lu_2_O_3_-iPSMA and ^177^Lu_2_O_3_-iFAP radionanosystems do not affect the integrity of liver cells and kidney function.

Figure 4 shows the [^18^F]FDG-micro-PET/CT images of mice after treatment (at 21 d) with ^177^Lu_2_O_3_ (SUV_TUMOR_ = 1.950), ^177^Lu_2_O_3_-iPSMA (SUV_TUMOR_ = 0.365), and ^177^Lu_2_O_3_-iFAP (SUV_TUMOR_ = 0.334), and an untreated mouse (control) (SUV_TUMOR_ = 2.847), in which the low metabolic activity in tumors treated with functionalized radionanoparticles can be observed. The significant difference between nonfunctionalized and functionalized radionanoparticles can be associated with the mechanisms mediated by receptors present in the tumor microenvironment that recognize iPSMA or iFAP ligands bound to the surface of the nanoparticles, leading to large tumor retention, high doses of radiation delivered to the tumor, and a further decrease in tumor metabolic activity.

The patient with multiple colorectal liver metastases (PSMA-positive; as determined by [^99m^Tc]Tc-iPSMA SPECT/CT imaging, received ^177^Lu_2_O_3_-iPSMA under a “compassionate use” protocol. The activity, administered intravenously, for the evaluation of the biokinetics and dosimetry in liver metastases, was 185 MBq (0.2 mg/0.8 mL). Figure 5 shows different molecular images of the patient: the [^18^F]FDG PET/CT images acquired on 27 August 2021, the [^99m^Tc]Tc-iPSMA SPECT/CT images at 3 h and 24 h post-injection (administration on 9 November 2021), and the ^177^Lu_2_O_3_-iPSMA SPECT/CT images acquired on 18 November 2021. A SPECT/CT axial section of the hepatic region is shown at the top of each image. As can be seen in Figure 5, the patient had multiple liver lesions with moderate PSMA expression, which successfully captured the ^177^Lu_2_O_3_-iPSMA nanosystem. It can also be observed that over almost three months, the metastases moved. This occurred due to the formation of several bilomas despite the fact that the patient had a prosthesis (stent) and two biliary catheters, because a necrotic tumor (approximately of 6 cm in diameter) obstructed the bile ducts at an earlier stage of the disease. The described condition of the patient’s liver can be appreciated in CT figures shown in Appendix A (Figure A1).

In Figure 5, the ^177^Lu_2_O_3_-iPSMA SPECT/CT image was contrast-enhanced to highlight the uptake of the radionanoparticles in most of the tumor lesions observed in the [^18^F]FDG PET/CT study as a proof of concept; however, the radionanosystem was also captured in the reticuloendothelial system of the lungs and, to a lesser extent, in the spleen, as observed in the detailed biokinetic behavior of the radiation source organs shown in Figure 6. The biodistribution in the patient does not correlate with that obtained in preclinical studies after intravenous administration of the radionanosystem (Figure 2). Differences in biodistribution patterns are attributed to the severe liver damage of the patient; the reason why there was a redistribution of radionanoparticle uptake by the reticuloendothelial system. The whole-body and abdominal SPECT images can be observed in Appendix A (Figure A2). It is important to emphasize that the biodistribution obtained in the patient in this research cannot be considered as the normal pattern for ^177^Lu_2_O_3_-iPSMA nanoparticles. The relevant point to be considered is that the radionanoparticle’s biodistribution will depend on the degree of disease progression.

Nevertheless, the biokinetic models obtained from the patient for each radiation source organ (Table 7) and metastatic lesions indicated that an administered activity of 1350 MBq would be safe and adequate to generate a total number of nuclear transformations sufficient to produce therapeutic radiation absorbed doses (from 42 to 202 Gy) in the liver metastases (Table 8), without exceeding maximum tolerated doses to any organ (Table 9). For example, it is well-known that with a maximum dose of 7 Gy to the lung parenchyma, after external irradiation (conventional radiotherapy), the probability of symptomatic pneumonia is 5% [22]. In the case presented in this research, the patient would receive 3.42 Gy per 1350 MBq (Table 8), but not homogeneously to the lung parenchyma, since the nanoparticles are trapped by the macrophages of the reticuloendothelial system; a behavior that also occurs in the spleen and liver. In other words, healthy lung parenchyma cells do not trap ^177^Lu_2_O_3_-iPSMA and are not a radiation source tissue; they only receive radiation from the nanoparticles trapped in the alveolar macrophages, which means that the actual radiation dose to healthy lung cells would be much lower than 3.42 Gy per 1350 MBq administered intravenously.

From Table 8, the total number of nuclear transformations that occur in liver metastases can be calculated ($\sum_{1}^{14}N=62.36\;\mathrm{MBq}\cdot h/\mathrm{MBq})$, which represents 72% of the total number of nuclear transformations calculated for the liver (86.59 MBq·h/MBq) (Table 7). Therefore, it is expected that the actual radiation dose to healthy liver cells would be less than 1 Gy per 1350 MBq administered intravenously. Although lungs and spleen doses are higher than that of the liver (Table 9), all of them are below the maximum tolerated radiation doses (MTD) recommended to the liver (MTD = 20–30 Gy), lungs (MTD = 20 Gy), and spleen (MTD = 15 Gy). Thus, for future applications, it would be necessary to develop specific dosimetric models for personalized endoradiotherapy of nanoparticles in patients.

Even though ^177^Lu_2_O_3_-iPSMA nanoparticles could be a viable therapeutic option for the patient evaluated in this research, it was not possible to follow up because she presented bowel obstruction due to intestinal adhesions formed as a consequence of the abdominal surgery for resection of the primary tumor. As a result, the patient had to undergo a second surgery, which complicated her condition until septic shock.

In the practice of targeted endoradiotherapy, it is necessary that the radiation absorbed dose calculations be performed as a personalized evaluation. In the case of ^177^Lu_2_O_3_-iPSMA application for the treatment of liver metastases, the custom dose calculation is even more critical, since the distribution of the radionanosystem can exhibit highly significant variations, depending on the patient’s liver damage. Although there is the option of administering radionanoparticles through the hepatic artery by an interventional radiology procedure to improve the distribution profile towards metastatic lesions, the intravenous route remains a less-invasive and less-expensive option, which is therefore, more feasible.

## 4. Discussion

As previously mentioned, ^177^Lu_2_O_3_-iPSMA biodistribution in the patient does not correlate with that obtained in preclinical studies after intravenous administration of the radionanosystem. However, it is expected that in most of the patients with liver neoplasms, ^177^Lu_2_O_3_-iPSMA nanoparticles follow the preclinical biodistribution pattern. This assertion is because the phenomenon of redistribution of radiocolloids, which involves lung uptake by induction of the pulmonary intravascular phagocytosis, occurs in patients with severe liver damage due to chronic biliary obstruction, extensive liver neoplasia, and liver infections [23,24]—three clinical conditions of the patient included in this research and not common for all patients. Therefore, the administration of ^177^Lu_2_O_3_-iPSMA nanoparticles to patients must be carried out under a strict personalized dosimetry procedure.

The PSMA protein is not only overexpressed in cancer cells. Although its higher expression is associated with advanced prostate cancer, PSMA is considered a multifunctional protein [12]. Among other functions, PSMA participates in the transduction of signals associated with angiogenesis and cell migration, which is why it is overexpressed in tumor neovasculature of various types of cancer, including metastatic colorectal cancer as shown in this research and in agreement with other authors [13,14]. ^177^Lu_2_O_3_-iPSMA nanoparticles could also be useful in the treatment of hepatocellular carcinoma, where PSMA is expressed both on the membrane of cancer cells (41% of tumors) and in the tumor neovasculature (90% of tumors) [16].

Interactions between nonmalignant and malignant cells form the tumor microenvironment. The tumor microenvironment provides essential elements involved in the processes of initiation and progression of cancer, and in the phases of metastasis and resistance to tumor therapy through intercellular communication, which is mainly regulated by dynamic networks and complex chemokines, cytokines, inflammatory enzymes, growth factors, and different components of the extracellular matrix [25]. As part of the cellular components of the tumor microenvironment, there are cancer-associated fibroblasts with high expression of FAP while PSMA expression is associated with the continuous angiogenic processes in the extracellular matrix. Therefore, the nanoplatforms containing ligands targeting FAP and PSMA, such as ^177^Lu_2_O_3_-iFAP, ^177^Lu_2_O_3_-iPSMA, or even hybrid ^177^Lu_2_O_3_-iFAP/iPSMA systems could improve tumor retention due to their specific binding to both fibroblasts and tumor neovasculature (Figure 7). It is expected that other polymeric and metallic nanoparticles labeled with ^177^Lu [26,27,28,29,30] show a behavior like that reported in this research, provided that they have the same size range and are functionalized with biomolecules directed to proteins of the tumor microenvironment.

It is important to mention that the ^177^Lu_2_O_3_-iPSMA nanoparticles developed in this research are not proposed as an option to replace prostate cancer therapies using ligands targeting PSMA, such as ^177^Lu-PSMA-617 or ^177^Lu-iPSMA [31,32]; what is proposed is the possibility of using ^177^Lu_2_O_3_-iPSMA for the treatment of liver metastases or tumors that express PSMA in the neovasculature, even with reduced or null expression of PSMA on the membrane of cancer cells. For example, Cuda et al. [14] reported that despite having observed PSMA expression in the neovasculature of colorectal liver metastases in ten patients when performing ^68^Ga-PSMA-11 imaging, none of them showed sufficient tumor avidity to be included in a ^177^Lu-PSMA-617 treatment protocol. It is precisely in these clinical cases that ^177^Lu_2_O_3_-iPSMA nanoparticles could be useful, since the first mechanism of tumor accumulation is the enhanced permeability and retention (EPR) effect, which initially takes advantage of the endothelial intercellular space size (from 400 nm up to 800 nm) of tumors, compared to 2 nm in normal endothelial cells (Figure 7). Once inside the tumor, ^177^Lu_2_O_3_-iPSMA nanoparticles are also retained by a specific mechanism mediated by the PSMA protein (Figure 7). The first tumor interaction of ^177^Lu_2_O_3_-iPSMA nanoparticles by the EPR effect, as well as their very fast uptake by the reticuloendothelial system, explains why other organs that express PSMA, such as the salivary glands, are not observed in the SPECT/CT image shown in this research (Figure A2).

Currently, ^90^Y-microspheres (size of 20–30 µm) are used for hepatic artery radioembolization in the treatment of liver metastases and hepatocarcinoma [33]. The radioactive microspheres clog the hepatic artery and act as a brachytherapy device [34]. The main advantage of the ^177^Lu_2_O_3_-iFAP and ^177^Lu_2_O_3_-iPSMA nanosystems is their targeted retention in the tumor microenvironment to deliver radiation doses as a localized and selective approach.

## 5. Conclusions

Preclinical results showed that ^177^Lu_2_O_3_-iPSMA and ^177^Lu_2_O_3_-iFAP inhibit colorectal HCT116 tumor progression due to prolonged tumor retention and a combination of ^177^Lu radiotherapy and iPSMA or iFAP molecular recognition. There were negligible uptake values in non-target tissues and no liver and renal toxicity evidence. The doses received by the patient’s liver metastases demonstrated the potential of ^177^Lu_2_O_3_-iPSMA for treating colorectal liver metastases. However, the biodistribution profile of radionanoparticles can exhibit highly significant variations, depending on the patient’s liver damage; the reason why personalized dosimetry calculations are crucial. Although, the intravenous administration of nanoparticles is less invasive and feasible, the option of administering ^177^Lu_2_O_3_-iPSMA and ^177^Lu_2_O_3_-iFAP through the hepatic artery using an interventional radiology procedure remains the next task to improve their distribution and dosimetry profile.

## Figures and Tables

**Figure 1 pharmaceutics-14-00720-f001:**
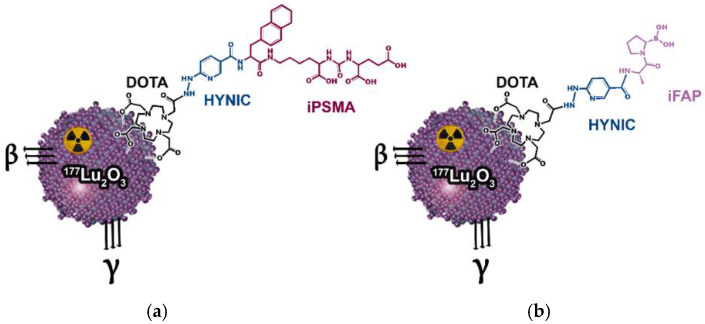
Schemes of the (**a**) ^177^Lu_2_O_3-_iPSMA and (**b**) ^177^Lu_2_O_3-_ iFAP nanosystems.

**Figure 2 pharmaceutics-14-00720-f002:**
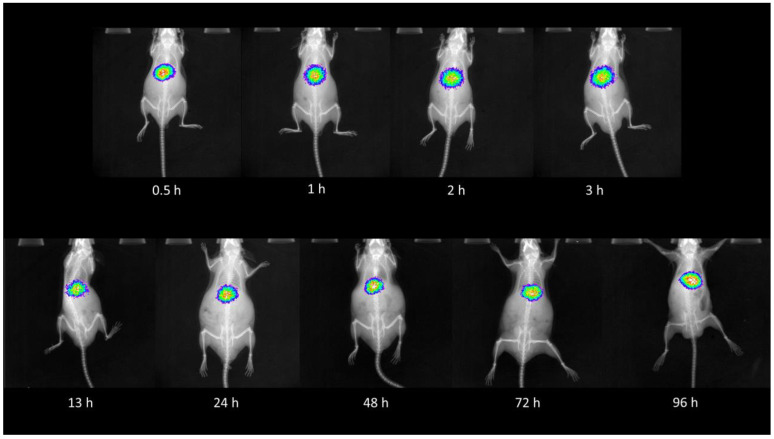
Radioluminescent images at various time points after the intravenous injection of ^177^Lu_2_O_3_-iPSMA (13 MBq, 0.1 mg) in healthy CD1 mice.

**Figure 3 pharmaceutics-14-00720-f003:**
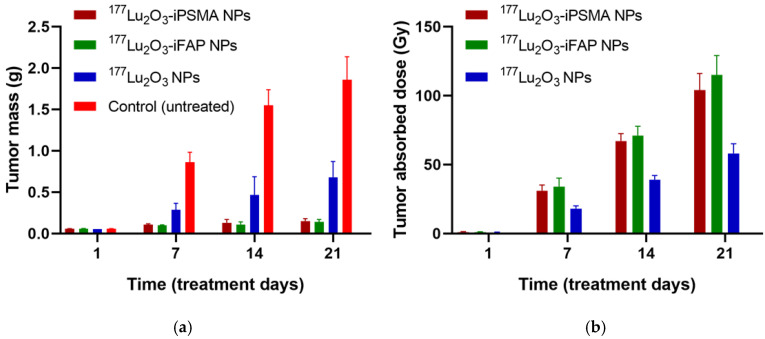
(**a**) Tumor size progression for ^177^Lu_2_O_3_-iPSMA, ^177^Lu_2_O_3_-iFAP, and ^177^Lu_2_O_3_ groups at different days of the treatment, without statistically significant difference (*p* > 0.05) between ^177^Lu_2_O_3_-iPSMA and ^177^Lu_2_O_3_-iFAP and with statistically significant difference (*p* < 0.05) of ^177^Lu_2_O_3_-iPSMA and ^177^Lu_2_O_3_-iFAP vs. ^177^Lu_2_O_3_ nanoparticles and the control group; (**b**) The mean radiation absorbed doses of ^177^Lu_2_O_3_-iPSMA, ^177^Lu_2_O_3_-iFAP, and ^177^Lu_2_O_3_ delivered to HCT116 tumors.

**Figure 4 pharmaceutics-14-00720-f004:**
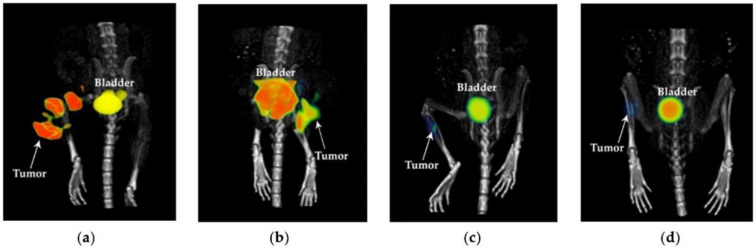
[^18^F]FDG-microPET/CT imaging of the (**a**) control (untreated mouse), (**b**) ^177^Lu_2_O_3_, (**c**) ^177^Lu_2_O_3_-iPSMA, (**d**) ^177^Lu_2_O_3_-iFAP groups at 21 days of treatment of mice bearing HCT116 tumors.

**Figure 5 pharmaceutics-14-00720-f005:**
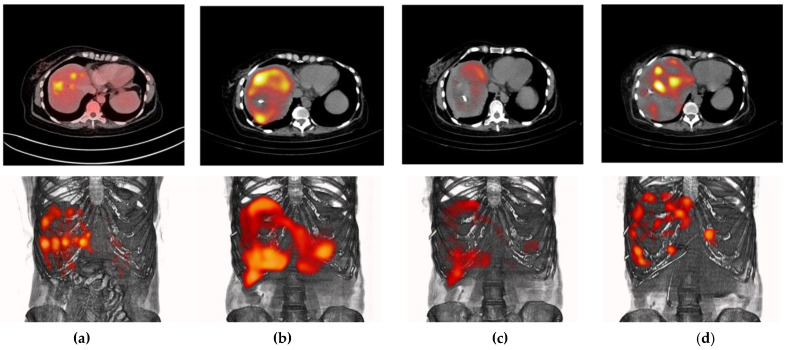
Different molecular images (top: axial sections; bottom: PET/CT or SPECT/CT images) of a patient with multiple colorectal liver metastases (PSMA-positive): (**a**) [^18^F]FDG PET/CT images acquired on 27 August 2021; [^99m^Tc]Tc-iPSMA SPECT/CT images at (**b**) 3 h and (**c**) 24 h post-injection (administration on 9 November 2021); and (**d**) ^177^Lu_2_O_3_-iPSMA SPECT/CT images acquired on 18 November 2021. It can also be observed that over almost three months, the metastases moved due to the appearance of bilomas and the presence of necrotic tumors. The ^177^Lu_2_O_3_-iPSMA SPECT/CT image was contrast-enhanced to highlight the uptake of the radionanoparticles in tumor lesions as proof of concept. The detailed biokinetic behavior of the source organs is shown in Figure 6.

**Figure 6 pharmaceutics-14-00720-f006:**
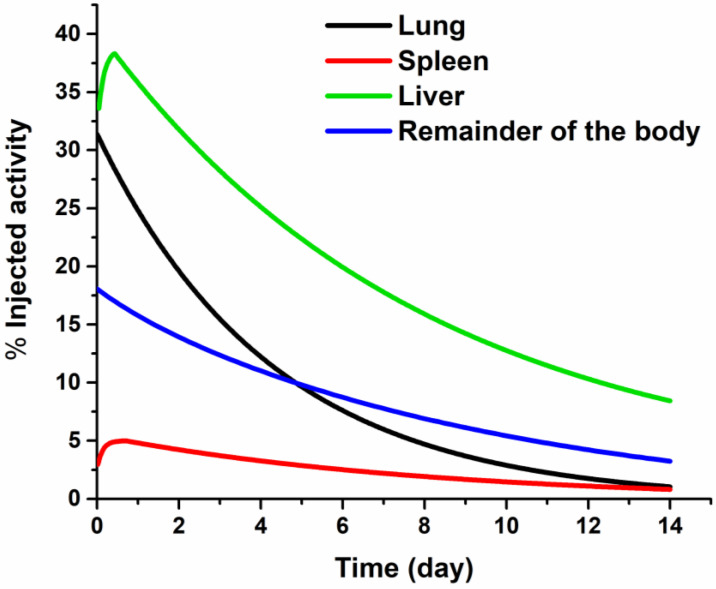
Biokinetic behavior of the source organs after the intravenous administration of ^177^Lu_2_O_3_-iPSMA nanoparticles in a patient with multiple colorectal liver metastases.

**Figure 7 pharmaceutics-14-00720-f007:**
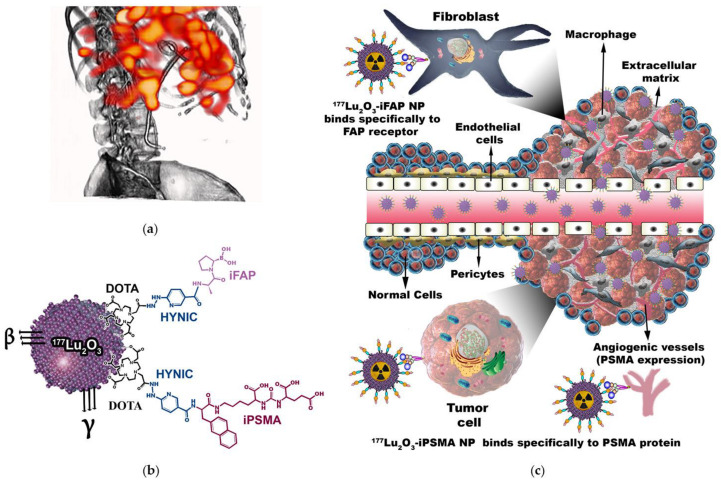
(**a**) Schematic “proof of concept” of a patient with colorectal cancer liver metastases with selective tumor uptake of (**b**) ^177^Lu_2_O_3_-iPSMA/^177^Lu_2_O_3_-iFAP nanoparticles (**c**) initially captured by the liver reticuloenthotelial system and by tumor metastases due to an enhanced permeability and retention (EPR) effect, but not by healthy hepatocytes of the liver parenchyma; subsequently, retained within tumors by a specific mechanism mediated both by overexpressed receptors on the surface of cancer cells (e.g., PSMA), and by receptors of the tumor microenvironment (e.g., FAP and PSMA).

**Table 1 pharmaceutics-14-00720-t001:** Biodistribution (% injected dose) of ^177^Lu_2_O_3_-iPSMA in CD1 mice at various time points (mean ± SD) (*n* = 3) after intravenous injection.

Organ	Time (h)			
	3	48	72	96
Heart	0.11 ± 0.03	0.08 ± 0.02	0.02 ± 0.01	0.01 ± 0.01
Liver	23.56 ± 2.13	18.08 ± 0.94	16.87 ± 1.37	14.59 ± 1.72
Lung	0.34 ± 0.32	0.21 ± 0.15	0.16 ± 0.09	0.12 ± 0.05
Pancreas	0.12 ± 0.04	0.04 ± 0.03	0.01 ± 0.01	0.00 ± 0.00
Spleen	1.61 ± 0.59	1.18 ± 0.91	0.99 ± 0.63	0.81 ± 0.35
Kidney	0.85 ± 0.11	0.41 ± 0.18	0.22 ± 0.07	0.19 ± 0.10
Brain	0.01 ± 0.01	0.00 ± 0.00	0.00 ± 0.00	0.00 ± 0.00

**Table 2 pharmaceutics-14-00720-t002:** Biodistribution (% injected dose) of ^177^Lu_2_O_3_-iFAP in CD1 mice at various time points (mean ± SD) (*n* = 3) after intravenous injection.

Organ	Time (h)			
	3	48	72	96
Heart	0.13 ± 0.03	0.05 ± 0.03	0.01 ± 0.01	0.01 ± 0.01
Liver	24.98 ± 1.84	18.93 ± 2.04	17.23 ± 1.41	15.01 ± 1.17
Lung	0.35 ± 0.09	0.20 ± 0.11	0.14 ± 0.04	0.08 ± 0.03
Pancreas	0.10 ± 0.07	0.02 ± 0.01	0.00 ± 0.00	0.00 ± 0.00
Spleen	1.52 ± 0.39	1.14 ± 0.49	0.79 ± 0.39	0.75 ± 0.28
Kidney	0.98 ± 0.10	0.52 ± 0.07	0.21 ± 0.07	0.18 ± 0.08
Brain	0.00 ± 0.00	0.00 ± 0.00	0.00 ± 0.00	0.00 ± 0.00

**Table 3 pharmaceutics-14-00720-t003:** Biodistribution (% injected dose) at various time points of ^177^Lu_2_O_3_-iPSMA in nude mice bearing HCT116 tumors (mean ± SD) (*n* = 3), after intratumoral injection.

Tissue	Time (h)			
	3	48	72	96
Heart	0.28 ± 0.11	0.19 ± 0.07	0.10 ± 0.08	0.07 ± 0.02
Lung	0.30 ± 0.10	0.21 ± 0.03	0.15 ± 0.04	0.11 ± 0.07
Liver	1.05 ± 0.38	0.94 ± 0.21	0.88 ± 0.12	0.84 ± 0.09
Spleen	0.39 ± 0.16	0.34 ± 0.10	0.31 ± 0.09	0.28 ± 0.07
Kidney	0.84 ± 0.19	0.63 ± 0.23	0.58 ± 0.17	0.39 ± 0.21
Tumor (%ID/g)	80.42 ± 5.87	79.32 ± 5.27	77.23 ± 6.01	75.98 ± 5.41

**Table 4 pharmaceutics-14-00720-t004:** Biodistribution (% injected dose) at various time points of ^177^Lu_2_O_3_-iFAP in nude mice bearing HCT116 tumors (mean ± SD) (*n* = 3), after intratumoral injection.

Tissue	Time (h)			
	3	48	72	96
Heart	0.22 ± 0.09	0.18 ± 0.09	0.14 ± 0.07	0.05 ± 0.02
Lung	0.26 ± 0.11	0.19 ± 08	0.16 ± 0.05	0.10 ± 0.04
Liver	1.18 ± 0.70	1.10 ± 0.42	0.83 ± 0.22	0.81 ± 0.13
Spleen	0.40 ± 0.14	0.35 ± 0.12	0.32 ± 0.11	0.26 ± 0.09
Kidney	0.91 ± 0.17	0.78 ± 0.15	0.62 ± 0.13	0.50 ± 0.18
Tumor (%ID/g)	83.87 ± 5.33	81.21 ± 5.64	78.58 ± 3.81	77.32 ± 4.25

**Table 5 pharmaceutics-14-00720-t005:** Biodistribution (% injected dose) at various time points of ^177^Lu_2_O_3_ in nude mice bearing HCT116 tumors (mean ± SD) (*n* = 3), after intratumoral injection.

Tissue	Time (h)			
	3	48	72	96
Heart	0.27 ± 0.06	0.21 ± 0.08	0.16 ± 0.07	0.11 ± 0.04
Lung	0.29 ± 0.13	0.25 ± 0.11	0.20 ± 0.14	0.18 ± 0.05
Liver	1.69 ± 0.97	1.40 ± 0.52	1.36 ± 0.45	1.29 ± 0.12
Spleen	0.55 ± 0.28	0.49 ± 0.25	0.40 ± 0.21	0.38 ± 0.14
Kidney	0.98 ± 0.31	0.95 ± 0.29	0.88 ± 0.43	0.83 ± 0.34
Tumor (%ID/g)	64.12 ± 4.98	57.25 ± 4.56	53.58 ± 3.25	49.52 ± 3.31

**Table 6 pharmaceutics-14-00720-t006:** Tumor metabolic activity ([^18^F]FDG/micro-PET; SUV(standard uptake value)), creatinine, aspartate aminotransferase (AST), alanine aminotransferase (ALT), and lactate dehydrogenase (LDH) serum concentrations in mice bearing HCT116 tumors after 21 days of treatment with ^177^Lu_2_O_3_-iPSMA, ^177^Lu_2_O_3_-iFAP, or ^177^Lu_2_O_3_ nanoparticles (mean ± standard deviation).

Treatment Group	SUV	Creatinine (mg/dL)	Aspartate Aminotransferase (AST) (IU/L)	Alanine Aminotransferase (ALT) (IU/L)	Lactate Dehydrogenase (LDH) (IU/L)
^177^Lu_2_O_3_-iPSMA	0.421 ± 0.092	0.208 ± 0.032	152 ± 13	71 ± 6	292 ± 37
^177^Lu_2_O_3_-iFAP	0.375 ± 0.104	0.197 ± 0.042	137 ± 16	65 ± 7	285 ± 44
^177^Lu_2_O_3_	1.821 ± 0.891	0.186 ± 0.051	142 ± 13	69 ± 5	294 ± 39
Control	2.654 ± 0.742	0.193 ± 0.061	148 ± 15	74 ± 8	302 ± 42

**Table 7 pharmaceutics-14-00720-t007:** Biokinetic models of ^177^Lu_2_O_3_-iPSMA nanoparticles, calculated from a patient with multiple colorectal liver metastases.

Organ	Biokinetic Model $A{\left(t\right)}_{VOI}$	$N=\overset{t=\infty }{\underset{t=0}{\int }}A{\left(t\right)}_{VOI}dt$ (MBq h/MBq)
Liver	$A{\left(t\right)}_{VOI}=-6.02e^{-0.1304t}+37.40e^{-0.0043t}+19.00e^{-2.5704t}$	86.59
Lung	$A{\left(t\right)}_{VOI}=31.40e^{-0.0073t}+0.14e^{-0.0043t}$	43.33
Spleen	$A{\left(t\right)}_{VOI}=-3.04e^{-0.2860t}+4.92e^{-7.1400t}+5.41e^{-0.0056t}$	9.56
Remainder of the body	$A{\left(t\right)}_{VOI}=17.60e^{-0.0054t}+0.22e^{-35.200t}$	32.60

**Table 8 pharmaceutics-14-00720-t008:** Total number of nuclear transformations (N) and radiation absorbed doses (Gy/1350 MBq) of ^177^Lu_2_O_3_-iPSMA nanoparticles in liver metastases, from a patient with metastatic colorectal cancer.

Liver Metastasis (Lesion Number)	Volume (cm^3^)	$N=\overset{t=\infty }{\underset{t=0}{\int }}A{\left(t\right)}_{VOI}dt$ (MBq h/MBq)	Dose (Gy) (per 1350 MBq)
L1	8.44	6.73	84.52
L2	6.12	3.96	68.32
L3	7.70	3.09	42.48
L4	9.26	4.23	48.38
L5	10.27	5.91	60.95
L6	3.01	2.76	95.87
L7	9.70	5.69	62.02
L8	3.47	2.59	78.10
L9	17.53	9.46	57.21
L10	2.58	2.70	109.10
L11	1.42	2.28	201.61
L12	7.76	5.22	70.66
L13	12.47	6.03	50.94
L14	2.15	1.71	82.29

**Table 9 pharmaceutics-14-00720-t009:** Radiation absorbed doses (Gy per 1350 MBq) of ^177^Lu_2_O_3_-iPSMA nanoparticles, calculated from a patient with multiple colorectal liver metastases.

Target Organ	Absorbed Doses
	(Gy)
Adrenals	1.18 × 10^−^ × 10^−01^
Brain	4.29 × 10^−02^
Breasts	7.05 × 10^−02^
Gallbladder wall	1.49 × 10^−01^
Lower large intestine wall	4.81 × 10^−02^
Small intestine	6.37 × 10^−02^
Stomach wall	9.79 × 10^−02^
Upper laege intestine wall	8.29 × 10^−02^
Heart	1.29 × 10^−01^
Kidneys	1.08 × 10^−01^
Liver	3.21 × 10^+00^
Lungs	3.42 × 10^+00^
Muscle	7.22 × 10^−^^02^
Ovaries	5.97 × 10^−^^02^
Pancreas	1.43 × 10^−^^01^
Red marrow	5.36 × 10^−^^02^
Osteogenic cells	1.94 × 10^−^^01^
Skin	5.81 × 10^−^^02^
Spleen	3.78 × 10^+00^
Thymus	7.52 × 10^−02^
Thyroid	5.86 × 10^−02^
Urinary bladder wall	5.61 × 10^−02^
Uterus	5.90 × 10^−02^
Total body	2.82 × 10^−01^

## Data Availability

Not applicable.

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
