# Peer review of "Targeted Endoradiotherapy with Lu2O3-iPSMA/-iFAP Nanoparticles Activated by Neutron Irradiation: Preclinical Evaluation and First Patient Image"

_pharmaceutics, 2022, doi:10.3390/pharmaceutics14040720_

Round 1

Reviewer 1 Report

The authors reported a concise way to prepare 177Lu-containing nanoparticles functionalized either with a PSMA or with a FAPI function alone or as hybrid particles. These particles were tested in vivo in mice to determine their biodistribution behavior. Additionally, a patient was chosen to evaluate the potential of the NPs in human, showing a high contribution to the “standard” 177Lu-PSMA-617 therapy.

The manuscript is nicely written and the experiments are well explained. In my mind the discussion about of the 177Lu-labeled NPs compared to the standard radiotracer 177Lu-PSMA-617 is too short and could be expanded.

Some minor comments:

Page 3, line 104: add space character after “2500”

Page 3, line 106: use correct nomenclature: [177Lu]LuCl3 and [177Lu]Lu2O3 (with “177” written superscript)

Page 4, line 149: Ah(t) “h” should be subscript

Page 6, line 247: “higher” instead of “greater”

Author Response

Dear Professor Garrigue,

Please find attached the answers, point-by-point, to the reviewers' comments.

Reviewer #1:

The authors reported a concise way to prepare 177Lu-containing nanoparticles functionalized either with a PSMA or with a FAPI function alone or as hybrid particles. These particles were tested in vivo in mice to determine their biodistribution behavior. Additionally, a patient was chosen to evaluate the potential of the NPs in human, showing a high contribution to the “standard” 177Lu-PSMA-617 therapy.

The manuscript is nicely written and the experiments are well explained. In my mind the discussion about of the 177Lu-labeled NPs compared to the standard radiotracer 177Lu-PSMA-617 is too short and could be expanded.

ANSWER:

 The authors appreciate the reviewer's positive comments on the research paper.

 In this research, 177Lu2O3-iPSMA and 177Lu2O3-iFAP nanoparticles targeting proteins overexpressed in the tumor microenvironment were informed: PSMA associated with angiogenesis (neovasculature) and FAP in cancer associated fibroblasts.

177Lu-PSMA-617 is successfully used for prostate cancer radiotherapy by targeting PSMA overexpressed on cancer cells. The authors consider that the most relevant differences between the nanoparticles and 177Lu-PSMA-617 were specifically discussed. Expanding the discussion trying to compare different radiopharmaceuticals aimed at different clinical conditions would not add more relevant aspects to the manuscript and could even be speculative.

Some minor comments:

Page 3, line 104: add space character after “2500”

ANSWER: It was corrected

Page 3, line 106: use correct nomenclature: [177Lu]LuCl3 and [177Lu]Lu2O3 (with “177” written superscript)

ANSWER: It was corrected

Page 4, line 149: Ah(t) “h” should be subscript

ANSWER: It was corrected

Page 6, line 247: “higher” instead of “greater”

ANSWER: It was corrected

Thank you very much for your comments, which have enriched our manuscript.

Reviewer 2 Report

This paper is a follow-up of previous studies carried out by the same group same on biological evaluation of Lu2O3-iPSMA/-iFAP nanoparticles for treating colorectal liver metastases. This is a well written paper that will be of interest to the readers of this journal. There is a good body of results that have been thoroughly interpretated. However, there are several major points that authors must address in order to be accepted for publication.

Major:

  1. An extensive review of the literature concerning 177Lu-labeled nanoparticles as theranostic radiopharmaceuticals for cancer must be added in Introduction Section.
  2. Authors avoid to make any correlations of the findings in Discussion Section with corresponding results of group’s previous paper and also analogous studies with 77Lu-PSMA-617, 77Lu-iPSMA and 68Ga-PSMA-11. In particular, in Discussion Section an extensive conversation and comparison of the results must be added, particularly in similar experimental in vitro & in vivo assays
  3. Authors must mention whether they tested the impact of NPs against HCT116 cells in terms of cytotoxicity, internalization, binding affinity) along with the results, as they did in their previous study  employing HepG2. These data are crucial in order to make a direct correlation of the compound’s activity towards these 2 types of cancer. This information could be added in Supporting.
  4. Authors must provide the in vitro stability studies of the compounds particularly in physiological conditions like PBS (pH 7.4). This study is necessary in order to procced to in vivo evaluation.
  5. In biodistribution studies the reported values are %ID/g or %ID/organ? Authors must include this information in Tables 1 and 2. Furthermore, the radioactivity in blood, muscle and in other organs (if available) should also be included in Tables 1, 2 and 3. Blood and urine samples must always be collected and measured in these type of studies. The percentage of radioactivity in a variety of organs and biological fluids like blood, urine are necessary in order to identify the main route of radioactivity excretion. Particularly in studies employing tumor bearing mice the tumor/blood ratio will verify the radioactivity retention in the tumor site. In Results Section a relevant comment about the main route of radioactivity excretion (such as via hepatobiliary system) in healthy and tumor bearing mice and the tumor/blood ratio must be included.
  6. By the end of the therapeutic protocol, did authors harvest and measure with a caliber the size of the tumors after the drug administration. An image along with the corresponding graph could de added in Supporting information.
  7. Figure 10 must be removed from the manuscript because it is based only in indications and not in solid proofs. For example, authors do not provide any substantial proof of selectivity between healthy and cancer hepatocytes. The effort of the authors to visualize a proposed mechanism of action is not supported by the existing data. Additional mechanistic/biochemical investigation of molecular pathways must alsotake place in order to safely establish the proposed action.

Minor:

  1. In Materials & Method, 2.1. Synthesis of lutetium nanoparticles Section must be lessened since it reports an already published method. Further details are not necessary.

Author Response

Dear Professor Garrigue,

Please find attached the answers, point-by-point, to the reviewers' comments.

Reviewer #2:

This paper is a follow-up of previous studies carried out by the same group same on biological evaluation of Lu2O3-iPSMA/-iFAP nanoparticles for treating colorectal liver metastases. This is a well written paper that will be of interest to the readers of this journal. There is a good body of results that have been thoroughly interpretated. However, there are several major points that authors must address in order to be accepted for publication.

Major:

An extensive review of the literature concerning 177Lu-labeled nanoparticles as theranostic radiopharmaceuticals for cancer must be added in Introduction Section.

ANSWER:

 The 177Lu2O3 nanoparticles evaluated in this research are not 177Lu‑labeled nanoparticles.

 177Lu2O3 are radioactive lutetium nanoparticles. Therefore, an extensive review of the literature concerning 177Lu-labeled nanoparticles as theranostic radiopharmaceuticals to be added in the introduction section of this manuscript is out of the research scope. The addition of an extensive review would confuse readers by diverting their attention from the central objective of our research work: “This research aimed to evaluate the biokinetics, dosimetry and preclinical therapeutic response of Lu2O3-iPSMA and Lu2O3-iFAP nanoparticles activated by neutron irradiation to demonstrate their potential for theranostic applications in nuclear medicine.”

 Nevertheless, in order to clarify the reviewer´s comment, the introduction section (second paragraph) was re-written as follows:

“Lutetium-177 has been widely used as the radionuclide of choice for peptide-targeted endoradiotherapy [5,6]. During the last decade many studies have reported the preclinical evaluation of polymeric and metallic nanoparticles labeled with 177Lu. However, the literature related to the preparation of lutetium oxide nanoparticles is scarce and focused on preparing luminescent doped systems (Lu2O3: Ln3+, Ln = Eu, Er, Yb) or contrast agents [7-10].” 

Authors avoid to make any correlations of the findings in Discussion Section with corresponding results of group’s previous paper and also analogous studies with 77Lu-PSMA-617, 77Lu-iPSMA and 68Ga-PSMA-11. In particular, in Discussion Section an extensive conversation and comparison of the results must be added, particularly in similar experimental in vitro & in vivo assays.

ANSWER: As novelty of this research, 177Lu2O3-iPSMA and 177Lu2O3-iFAP nanoparticles were used to target proteins overexpressed in the tumor microenvironment of liver metastases or hepatocarcinomas: PSMA associated to angiogenesis (neovasculature), and FAP on fibroblast associated to cancer.

 177Lu-PSMA-617, 177Lu-iPSMA and 68Ga-PSMA-11 are radiopharmaceuticals used for PSMA imaging (68Ga-PSMA-11) or for prostate cancer radiotherapy by targeting PSMA overexpressed on cancer cells (177Lu-PSMA-617, 177Lu-iPSMA). Therefore, 177Lu-PSMA-617, 177Lu-iPSMA and 68Ga-PSMA-11 are not analogous of 177Lu2O3-iPSMA and 177Lu2O3-iFAP nanoparticles, and the addition of “an extensive conversation and comparison of the results, particularly in similar experimental in vitro & in vivo assays” suggested by the reviewer is not justified.

  1. Authors must mention whether they tested the impact of NPs against HCT116 cells in terms of cytotoxicity, internalization, binding affinity) along with the results, as they did in their previous study employing HepG2. These data are crucial in order to make a direct correlation of the compound’s activity towards these 2 types of cancer. This information could be added in Supporting.

ANSWER:

 As mentioned in the introduction section (page 2, fourth paragraph), PSMA is overexpressed in the neovasculature (tumor microenviroment) of hepatocarcinomas.

 Prostate-specific membrane antigen (PSMA) is a protein that is overexpressed in 90% of advanced prostate tumors and the neovasculature of the tumor microenvironment of many types of advanced cancer, such as metastatic colon cancer, triple-negative breast cancer, and hepatocarcinoma, among others [12-16].”

 177Lu2O3-iPSMA and 177Lu2O3-iFAP reported in this research were used to target proteins overexpressed in the tumor microenvironment of liver metastases or hepatocarcinomas: PSMA associated to angiogenesis (neovasculature), and FAP on fibroblast associated to cancer.

 Studies regarding in vitro recognition of 177Lu2O3-iPSMA and 177Lu2O3-iFAP by their respective target proteins have already been published (Page 2, sixth paragraph of the introduction section):

 “The 177Lu2O3-iPSMA nanosystem demonstrated in vivo stability and affinity (Kd=5.7 nM) for the PSMA protein, while 177Lu2O3-iFAP showed significant membrane binding in stromal cells expressing FAP receptors [17,19].”

 It is clear that our target was not the protein overexpressed on the surface of HCT116 cells. Therefore, studies suggested by the reviewer on binding affinity, internalization, etc. are not justified.

 Authors must provide the in vitro stability studies of the compounds particularly in physiological conditions like PBS (pH 7.4). This study is necessary in order to procced to in vivo evaluation.

ANSWER:

 In vitro and in vivo stability of 177Lu2O3 have already been demonstrated [17]. Studies in PBS (pH 4) cannot be considered as a proof of in vitro stability.

In biodistribution studies the reported values are %ID/g or %ID/organ? Authors must include this information in Tables 1 and 2. Furthermore, the radioactivity in blood, muscle and in other organs (if available) should also be included in Tables 1, 2 and 3. Blood and urine samples must always be collected and measured in these type of studies. The percentage of radioactivity in a variety of organs and biological fluids like blood, urine are necessary in order to identify the main route of radioactivity excretion. Particularly in studies employing tumor bearing mice the tumor/blood ratio will verify the radioactivity retention in the tumor site. In Results Section a relevant comment about the main route of radioactivity excretion (such as via hepatobiliary system) in healthy and tumor bearing mice and the tumor/blood ratio must be included.

 ANSWER:

The objective of the preclinical study in mice was to establish the organs of radiotracer accumulation (source organs) and the elimination pathways of a therapeutic radiopharmaceutical. Tumor/blood ratios and %ID/g are critical parameters in the development of radiopharmaceuticals for molecular imaging (diagnostic purposes). 

In the development of therapeutic radiopharmaceuticals, the %ID in source organs must be used (accumulation of radioactivity as a function of time in the whole organ), which is mandatory for preclinical dosimetric calculations. For preclinical dosimetry, the use of %ID/g is a mistake. In our research, only the tumor mass was expressed as %ID/g in order to normalize a tumor mass value in all mice groups.

By the end of the therapeutic protocol, did authors harvest and measure with a caliber the size of the tumors after the drug administration. An image along with the corresponding graph could de added in Supporting information.

ANSWER:

 The graph of tumor size progression (from day 1 to day 21) is shown in Figure 3; and Figure 4 shows tumor imaging of the mice at the end of each treatment.

Figure 10 must be removed from the manuscript because it is based only in indications and not in solid proofs. For example, authors do not provide any substantial proof of selectivity between healthy and cancer hepatocytes. The effort of the authors to visualize a proposed mechanism of action is not supported by the existing data. Additional mechanistic/biochemical investigation of molecular pathways must alsotake place in order to safely establish the proposed action.

 ANSWER:

 The authors consider that it is very important to keep Figure 10, as it is a crucial tool to clarify various aspects mentioned in the discussion section. Fig. 10 is not used as a result, it is used as a useful scheme in understanding the discussion and conclusions of the manuscript.

Minor:

  1. In Materials & Method, 2.1. Synthesis of lutetium nanoparticles Section must be lessened since it reports an already published method. Further details are not necessary.

ANSWER:

The authors consider that it is relevant a brief description of the Lu2O3 nanoparticle synthesis. Therefore,  the “Materials & Method, 2.1. Synthesis of lutetium nanoparticles” section should remain.

Thank you very much for your comments

Reviewer 3 Report

PSMA and iFAP ligands to specifically target PSMA and FAP proteins is an emerging radiopharmaceutical tool in imaging.

The use of the same biochemical path, using Lu-177 beta emissions and nanoparticle carriers for improved targeting is an innovative approach. The authors aimed to evaluate the dosimetry and therapeutic response of Lu2O3-iPSMA and Lu2O3-iFAP nanoparticles activated by neutron irradiation in preclinical tumor tissues. Also, a patient with multiple colorectal liver metastases (PSMA-positive) received 177Lu2O3-iPSMA under a “compassionate use” protocol.

The presented results are well sustained by the experimental data and clear protocols. The potential of 177Lu2O3-iPSMA for treating colorectal liver metastases is to be further investigated and demonstrated; still, these results are encouraging and data are valuable for the scientific community.

Author Response

Reviewer #3:

The use of the same biochemical path, using Lu-177 beta emissions and nanoparticle carriers for improved targeting is an innovative approach. The authors aimed to evaluate the dosimetry and therapeutic response of Lu2O3-iPSMA and Lu2O3-iFAP nanoparticles activated by neutron irradiation in preclinical tumor tissues. Also, a patient with multiple colorectal liver metastases (PSMA-positive) received 177Lu2O3-iPSMA under a “compassionate use” protocol.

The presented results are well sustained by the experimental data and clear protocols. The potential of 177Lu2O3-iPSMA for treating colorectal liver metastases is to be further investigated and demonstrated; still, these results are encouraging and data are valuable for the scientific community

ANSWER:

 The authors appreciate the reviewer's positive comments on the research paper

Round 2

Reviewer 2 Report

Even though authors do not address some issues i pointed out, I think that it can be accepted for publication.

Author Response

The authors appreciate the reviewer's comments.